# A Compact Aperture-Sharing Sub-6 GHz/Millimeter-Wave Dual-Band Antenna

**DOI:** 10.3390/s23094400

**Published:** 2023-04-30

**Authors:** Qinghu Zhang, Bitian Chai, Jianxin Chen, Wenwen Yang

**Affiliations:** 1School of Information Science and Technology, Nantong University, Nantong 226019, China; 2The Nantong Research Institute for Advanced Communication Technologies, Chongchuan District, Nantong 226019, China

**Keywords:** steerable beam, dual polarization, millimeter-wave MIMO, aperture-sharing

## Abstract

In this article, a microwave (MW)/millimeter wave (MMW) aperture-sharing antenna is proposed. The antenna is constructed using two orthogonal columns of grounded vias from a 3.5 GHz slot-loaded half-mode substrate-integrated waveguide (HMSIW) antenna. These vias are reused to create two sets of 1 × 4 MMW substrate-integrated dielectric resonator antenna (SIDRA) arrays. With this proposed partial structure reuse strategy, the MW antenna and MMW arrays can be integrated in a shared-aperture manner, improving space utilization and enabling dual-polarized beam steering capability in the MMW band, which is highly desirable for multiple-input multipleoutput (MIMO) applications. The integrated antenna prototype was manufactured and measured for verification. The 3.5 GHz antenna has a relative bandwidth of 3.4% (3.44–3.56 GHz) with a peak antenna gain of 5.34 dBi, and the 28 GHz antenna arrays cover the frequency range of 26.5–29.8 GHz (11.8%) and attain a measured peak antenna gain of 11.0 dBi. Specifically, the 28 GHz antenna arrays can realize dual-polarization and ±45° beam steering capability. The dual-band antenna has a very compact structure, and it is applicable for 5G mobile communication terminals.

## 1. Introduction

New spectrum resources have been introduced in 5G to increase data capacity, mainly including the sub-6 GHz (450 MHz–6 GHz) and MMW bands (24.25–52.6 GHz) [1,2,3]. The deployment of new spectra will undoubtedly lead to an increase in antenna components in mobile terminals. Since mobile terminals already have very limited internal spaces, designing new terminal antennas will present significant challenges [4,5]. In earlier studies, the structure of adjacent placement of high- and low-frequency antennas was proposed [6]. However, this scheme has extremely low space utilization and cannot effectively reduce the size of the integrated antenna. To improve space utilization, shared-aperture antenna technology has been proposed, and it is considered to be a very promising technique. For a MW/MMW aperture shared antenna, the size of its MW antenna part should be very compact to fit the demand of terminal use. On the other hand, the MMW arrays with steering beams and dual polarization are highly desirable for the applications of MIMO due to their high gain, wide angle coverage (from steering beams), and enhanced channel capacity (from dual polarization) [7,8,9].

Numerous novel and creative shared-aperture designs using various technologies have been proposed. The first category is the stacked structure [10,11,12,13]. In [10], a high-frequency magneto-electric dipole antenna was placed above a low-frequency patch antenna. The second category is the embedded structure, in which the high-frequency antenna is embedded into the low-frequency antenna [14,15,16,17,18,19,20,21,22]. For instance, in [17], the MMW DRA was embedded in the MW patch antenna in a co-planar form. The mode composite transmission line structure can be used for constructing MW/MMW shared-aperture antennas. With this structure, the combinations of MW microstrip patch array and MMW SIW slot array antenna, MW patch and MMW SIW slot antenna, as well as MW patch antenna and MMW slotted cavity antenna were proposed in [23,24,25], respectively, to realize shared-aperture antennas. The reuse of metasurface is also proposed to achieve the aperture integration of multi-frequency antennas [26,27]. In [27], the S-band radiating surface was reused as the frequency selective surface (FSS) of the cavity-backed slot antenna array operating at K-band. Furthermore, some other ingenious approaches were also investigated, such as the shared-aperture integration of MMW transmit/reflect arrays and various MW antennas [28,29,30], as well as MMW Fabry-Perot cavity (FPC) antenna and various MW antennas [31,32]. For example in [29], the dual-band operation is realized through the combination of MMW reflectarray antenna and MW patch antenna array. In [32], the shared-aperture antenna is constructed by integrating MMW FPC antenna into sub-6 GHz patch antenna. It is worth noting that the isolation between the two frequency bands is also studied in this literature. For some designs, such as [14,22,24], cavity-backed antennas and SIW antennas which have natural high-pass characteristics are chosen as MMW antenna schemes to achieve high isolation between MW and MMW bands. In addition, the filter structures can be introduced in the feed lines to improve the isolation; for example, in [6], a low-pass filter is introduced to block the transmission of high-frequency signals.

In general, the current shared-aperture antennas still have some flaws: (1) the antenna sizes are large, making them difficult to be applied to the mobile terminals; and (2) most designs cannot support dual-polarized and wide-angle beam scanning in the MMW band.

In light of these issues, this article suggests a 3.5 and 28 GHz aperture-sharing antenna that is based on partial structure reuse strategy. Figure 1a–c depict the concept of the antenna. First, the HMSIW antenna is selected as the MW antenna due to its compact size. It is built on a substrate with high permittivity for further miniaturization and easier construction of MMW SIDRAs. The two orthogonal columns of grounded vias of the HMSIW antenna are then reused as two sets of 1 × 4 MMW SIDRA arrays, which are *x*-polarized and *y*-polarized arrays, respectively [33]. Since the array element spacing is roughly maintained at 0.5*λ*_02_ (*λ*_02_ is the wavelength in free space at 28 GHz) and each element can be fed independently, the MMW arrays can support ±45° beam steering [34]. To expand the bandwidth of the MW antenna, the slot-loading technique is further used to shift the higher-order mode downward to combine with the fundamental mode. Since the SIDRA antenna has natural high-pass characteristics, high isolation can be achieved between the two bands. The proposed antenna might have a use in terminals such as illustrated in Figure 1d.

## 2. Geometry

Figure 2a depicts the three-dimensional overview of the presented aperture-sharing antenna, which consists of two layers of Sub1 and Sub2. Sub1 is made of Rogers RT 6006 material with a permittivity *ε_r_*_1_ = 6.15 and a loss tangent of 0.002, while Sub2 is made of Rogers 4003C with *ε_r_*_2_ = 3.55 and tan*δ* = 0.0027. The coaxial probe numbered #1 excites the 3.5 GHz HMSIW antenna. The inner conductor of the probe is inserted into Via1, while the outer conductor is soldered to the ground. Figure 2b illustrates the upper surface of the HMSIW antenna, where two pairs of slots (Slot1 and Slot2) have been etched to enhance the bandwidth. The HMSIW antenna’s two orthogonal columns of grounded vias are reused as two sets of 1 × 4 28 GHz SIDRA arrays. The two linear arrays are evenly arranged along the *x*- and *y*-directions, and the element interval is 0.51*λ*_02_. Two sets of 1 × 4 “H-shaped” slots, which are used to feed the arrays with the microstrip line ports numbered #2~#9, are etched on the ground plane. Proper array spacing and the individual feed of each element enable the wide scanning angles of ±45° for the MMW arrays. Figure 2c presents the exploded view of the SIDRA element. Two types of grooves are constructed in Sub1: the internal non-metalized grooves are drilled to build the SIDRA, while the external metalized grooves are connected with the upper surface and ground plane, constructing a metallic cavity for the MMW antenna as well as a grounded via for the MW antenna.

All the antenna parameters are listed in the caption of Figure 2, and the following simulation results are obtained with HFSS 18.9 [35].

## 3. Antenna Design

Since the proposed partial structure reuse strategy results in very little interaction between MW and MMW antennas, the two parts can be designed separately.

### 3.1. Dual-Mode HMSIW MW Antenna at 3.5 GHz

The 3.5 GHz MW antenna design can be considered first. The preliminary dimension of the antenna can be calculated according to the classical SIW antenna theory [36,37] by setting the dominant mode of the TM_11_ mode to resonate at 3.5 GHz for potential 5G applications. Figure 3 shows the electric field distribution corresponding to TM_11_, TM_22_, and TM_33_ modes of the full-mode SIW (FMSIW) resonant cavity at 3.5 GHz, 7.4 GHz, and 8.2 GHz. The electric field in the FMSIW resonant cavity is symmetrical at the central plane along the *x*-direction in Figure 3. Hence, the plane can be equivalent to a virtual magnetic wall. If the FMSIW is divided into two halves along the virtual magnetic wall, each half is called the HMSIW, and it can support almost half of the original electric field distribution. Figure 4a presents the initial model of the proposed HMSIW antenna. Figure 4b illustrates the corresponding |S_11_|, which reveals three resonant modes in the 3–9 GHz frequency range. Figure 5 shows the electric field amplitude distributions of the HWSIW. When comparing with the field distribution of FMSIW modes, it can be concluded that the three modes correspond to TM_11_ mode at 3.5 GHz, TM_22_ mode at 7.1 GHz, and TM_33_ mode at 8 GHz, respectively.

The operating bandwidth of the TM_11_ mode is thoroughly narrow because of the compact size of the antenna. The idea is to shift the higher-order mode TM_22_ mode downward to combine with the TM_11_ mode in order to increase its bandwidth. The slot-loading technique is used in the design to implement this idea. Figure 6 presents the current distributions of the two modes to help investigate the slot-loading principle. The current distribution reveals that when Slot1 is loaded, the TM_11_ mode will not be affected since the current is parallel to the slot, but the TM_22_ will be significantly affected (shifted down) as the current is cut by the slot (current path will be increased). Similarly, if Slot2 is further introduced, the TM_22_ mode will continuously be affected because the current is still cut by the slot, and the TM_11_ mode will also be slightly affected since a small amount of current of this mode is cut as well.

A parametric study of the slot lengths is conducted to clearly verify the validity of the proposed slot-loading technique. With reference to Figure 7a, we can recognize that the frequency of the TM_22_ mode decreases rapidly while the frequency of TM_11_ roughly remains constant with the increase in length (*L_g_*). As can be learned through Figure 7b, with the increase in the length of Slot2 (*W_g_*), the frequency of the TM_11_ mode slightly shifts down and the frequency of TM_22_ continues to decrease quickly. Finally, the two modes can be merged, broadening the bandwidth from 1.3% to 3.4%.

### 3.2. SIDRA at 28 GHz Band

The design of SIDRA has been well presented in [38]. Figure 8a shows the simulated |S_11_| of a single SIDRA element and the gain of a 1 × 4 array. The proposed slot-feed SIDRA structure has two resonances, which are the lower-frequency resonance from the DRA mode and the higher-frequency resonance from the feeding slot mode. The element has a relative impedance bandwidth of 12.5% (26.1–29.6 GHz), while the peak antenna gain of the array is 11.5 dBi. Figure 8b shows the electrical field distribution of the internal reference surface at 27 GHz of the DRA, and it follows the ${\mathrm{TE}}_{111}^{x}$ mode distribution. Parametric studies were conducted to further investigate the two resonant modes. As shown in Figure 9a, as increasing the thickness *h*_2_ of the substrate, the lower resonant mode shifts down while the higher one stays stable, which indicates that the lower resonance is affected mainly by the DRA mode. Figure 9b illustrates that when the length of H-shaped slot *L_s_* increases, the higher resonant frequency shifts down while the lower resonant frequency remains almost fixed. This confirms that the higher resonance is influenced mainly by the slot mode. 

The two sets of 1 × 4 SIDRA arrays can steer the beams along *x*-axis polarization and *y*-axis polarization, respectively. Here, a 1 × 4 array along the *x*-axis is taken as an example to demonstrate the beam scanning characteristic. Figure 10a shows the two-dimensional diagram of the beam scanning array. The relative phase shift of *ψ*, 2*ψ,* and 3*ψ* are implemented in the elements. The radiation angle *θ* can be calculated by
(1)$$\theta =\sin^{-1}\text{·}(\frac{\psi c}{\omega_{RF}d})$$
where *c* is the velocity of light, $\omega_{RF}$ is the center RF frequency of the system, and *d* is the interval of the elements, which equals 0.51 *λ*_02_. The required phase shift *ψ* to generate the given beam position *θ* is calculated by Equation (1), and the relative phase shifts of *ψ*, 2*ψ,* and 3*ψ* are assigned to each excitation port [39]. Figure 10b shows the simulated steering beams at 28 GHz. It can be observed that the MMW SIDRA array has the ability to achieve beam scanning angles from −45° to +45° based on the standard of 3-dB scanning loss and −5-dB sidelobe level [7].

### 3.3. Design Guideline

Based on the above analysis, a brief design guideline can be concluded. First, a substrate with high permittivity (ε_r1_ > 6) should be chosen due to the need for a compact size of MW antenna and the construction of MMW DRA. Considering the co-aperture of MW and MMW antennas, the thickness of the substrate should be balanced with the planar size of the MMW antenna. Then, several design steps are given as follows:

Determining the initial size of the MW antenna according to the following empirical formula: $L_{siw}$ ≈ 0.5*λ_g_*, *λ_g_* = *λ*_01_/$\sqrt{\varepsilon_{eff}}$, where $\varepsilon_{eff}$= (*ε_r_*_1_ + 1)/2 and *λ*_01_ is the wavelength in the vacuum at 3.5 GHz [36,37].

Determining the initial size of the MMW antenna. The initial calculation can be performed with the classical dielectric waveguide model (DWM) for the SIDRA [40]. The size of the cavity *L_m_* can be set to be around *λ*_02_/2.

Replacing the grounding vias of the MW HMSIW antenna with MMW SIDRA arrays and introducing the slot-loading technique to expand the MW impedance bandwidth.

Optimizing the final structure to achieve good impedance and radiation performance.

### 3.4. Overall Integrated Structure Analysis

The two antennas can be combined to form a shared-aperture antenna once the independent designs of MW and MMW antennas are complete. To testify the coupling between the two different bands and the two polarizations in the MMW band, the S-parameters between different feeding ports should be investigated. As is shown in Figure 11a, the isolations between the MMW ports (ports 2–5) and MW port (port 1) are more than 50 dB in the 3.5 GHz frequency band and more than 20 dB in the 28 GHz frequency band. In Figure 11b, it is illustrated that the isolations between the two polarizations in the MMW band are better than 25 dB.

## 4. Simulation and Measurement Verification

To verify the viability of the presented concept, the co-designed antenna was fabricated as shown in Figure 12. The upper substrate (Sub1) and lower substrate (Sub2) are made, respectively, and tightly fixed with several M2 screws. A one-quarter Wilkinson power divider is introduced for one 1 × 4 MMW antenna array to measure the radiation patterns at a 0° scanning angle. The power divider consists of three equally divided power dividers, which convert the input impedance at the antenna unit to *Z*_0_ = 50 Ω through a *λ_g_* impedance converter. The function of a resistor is to improve the isolation of the output port of the power divider, with resistance R = 2 × *Z*_0_. The model of the resistor is a 100 Ω Panasonic resistor packaged as 0402.

Figure 13 illustrates the schematic of the measurement setup for testing the radiation patterns. The measurement was conducted in an anechoic chamber. An Agilent E8257D signal generator generated the input RF signal of the standard gain horn. The proposed antenna under test was used as the receiving end, and we used the spectrum analyzer (Agilent E4447A) to monitor the signal spectrum.

### 4.1. Measurement Results of 3.5 GHz MW Antenna

Figure 14a depicts the measured, simulated gains and |S_11_|s in the MW band. The measured impedance bandwidth is 3.4%, covering 3.44–3.56 GHz. The measured and simulated peak gains are 5.34 dBi and 5.72 dBi. As described in Figure 15, the measured and simulated results of the radiation pattern at 3.46 GHz and 3.53 GHz are consistent with each other. The measured cross-polarizations are better than −17.8 dB within the 3 dB beamwidth. The measured front-to-back ratio is greater than 11 dB.

### 4.2. Measurement Results of 28 GHz MMW Antenna

The measured, simulated gains and |S_11_|s for the MMW array are plotted in Figure 13b. The measured impedance bandwidth is 11.8% (26.5 GHz–29.8 GHz). Within the bandwidth, the simulated gain ranges between 9.7 and 11.5 dBi, while the measured gain is between 9.2 and 11.0 dBi. Figure 16 depicts the radiation patterns at 28 GHz of the MMW array at a 0° scanning angle. At the 28 GHz frequency band, the sidelobe level of the H-plane is better than −12.8 dB. The cross-polarization level is better than −21 dB.

### 4.3. Comparison

Table 1 provides a comprehensive comparison between the presented antenna and the reported aperture-sharing antennas. As can be seen from the table, the designs in [9,10,24,28,29,30] cannot support MMW beam scanning, greatly limiting their applications. Although dual-polarization and beam scanning for the MMW band are both possible in [17,20], the antenna in [17] is an end-fire design, whereas the antenna in [20] has a much larger volume. Compared to the reported designs, the proposed antenna has a more compact size. More importantly, it can simultaneously support wide-angle beam scanning and dual-polarization for the MMW band.

## 5. Conclusions

This article proposed, simulated, and measured a compact aperture-sharing antenna operating at 3.5 and 28 GHz bands with dual-polarization and beam steering in the MMW frequency band. The slot-loading technology is introduced in the 3.5 GHz HMSIW antenna design to broaden the bandwidth with mode analysis and parametric study conducted. The reuse of the orthogonally arranged grounded vias of the HMSIW antenna enables the dual-polarized beam steering ability for 28 GHz SIDRA arrays. The measurement results indicate that the MW and MMW antennas can work independently. The proposed shared-aperture antenna could be an attractive competitor for wireless communication terminal applications.

## Figures and Tables

**Figure 1 sensors-23-04400-f001:**
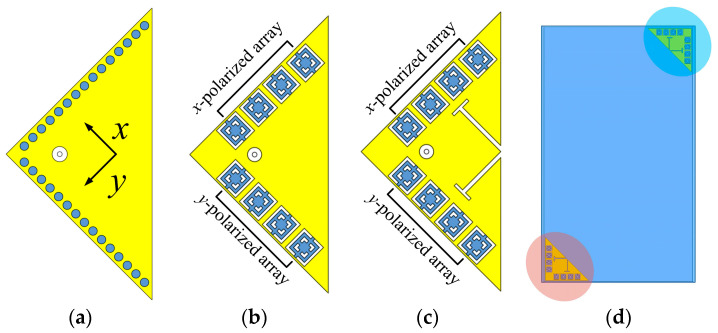
(**a**) HMSIW antenna. (**b**) Grounded vias of the HMSIW antenna are reused as SIDRA arrays. (**c**) Slot-loading technique is used to enhance the bandwidth of HMSIW antenna. (**d**) Potential application in terminals.

**Figure 2 sensors-23-04400-f002:**
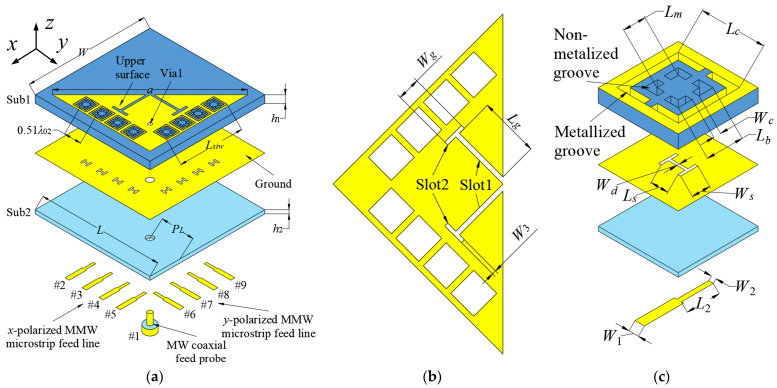
Configuration of the proposed antenna. (**a**) 3D view. (**b**) The upper surface of 3.5 GHz MW antenna. (**c**) The 28 GHz SIDRA element. (Design parameters: *W* = *L* = 40, *a* = 48, *L_siw_* = 21.1, *P_L_* = 11.9, *h*_1_ = 2.54, *h*_2_ = 0.305, *L_g_* = 8.8, *W_g_* = 3.8, *W*_3_ = 0.7, *L_c_* = 4.6, *L_m_* = 1.6, *L_b_* = 2.6, *W_c_* = 0.5, *W_d_* = 0.2, *L_s_* = 1.4, *W_s_* = 1.25, *W*_1_ = 0.7, *L*_2_ = 2.4, *W*_2_ = 0.35. Units: mm).

**Figure 3 sensors-23-04400-f003:**
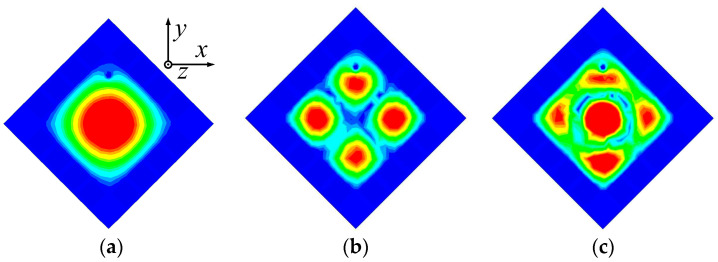
Electrical field distribution at different resonant modes of the FMSIW antenna. (**a**) TM_11_ mode at 3.5 GHz. (**b**) TM_22_ mode at 7.4 GHz. (**c**) TM_33_ mode at 8.2 GHz.

**Figure 4 sensors-23-04400-f004:**
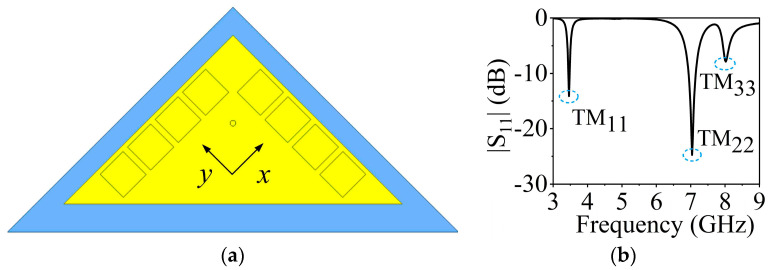
(**a**) Initial model of the HMSIW antenna. (**b**) |S_11_| of the HMSIW antenna.

**Figure 5 sensors-23-04400-f005:**
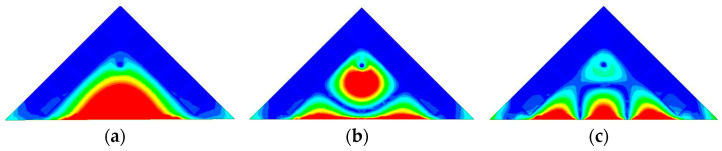
Electrical field distribution at different resonant modes of the HMSIW antenna. (**a**) TM_11_ mode at 3.46 GHz. (**b**) TM_22_ mode at 7.10 GHz. (**c**) TM_33_ mode at 8.00 GHz.

**Figure 6 sensors-23-04400-f006:**
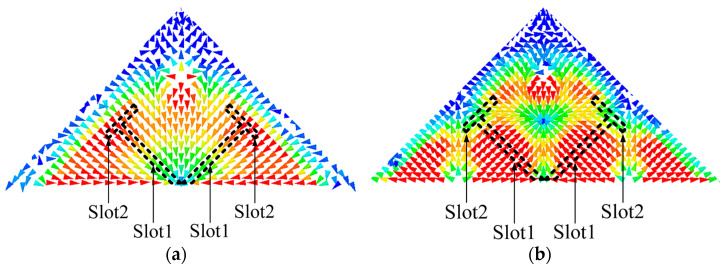
(**a**) The current distribution of TM_11_ mode with envisioned slots. (**b**) The current distribution of TM_22_ mode with envisioned slots.

**Figure 7 sensors-23-04400-f007:**
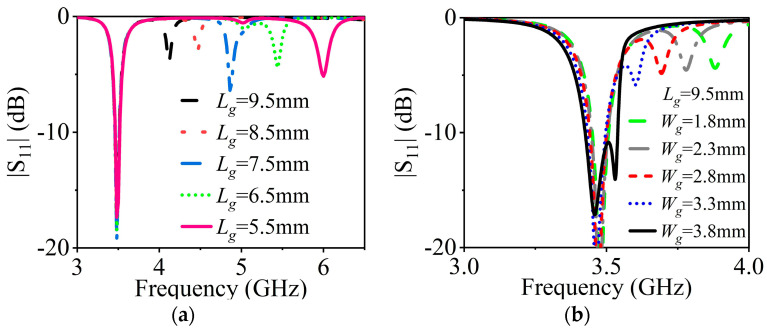
(**a**) Influence of the length Lg of Slot1 on |S_11_|. (**b**) Influence of the length *W_g_* of Slot2 on |S_11_| when *L_g_* = 9.5 mm.

**Figure 8 sensors-23-04400-f008:**
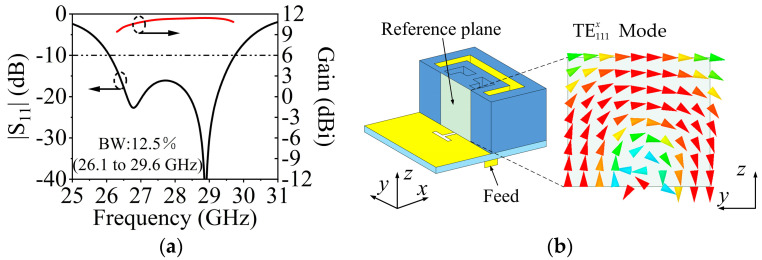
(**a**) Simulated |S_11_| and gain of the SIDRA. (**b**) Simulated *E*-field distribution inside the SIDRA at 27 GHz.

**Figure 9 sensors-23-04400-f009:**
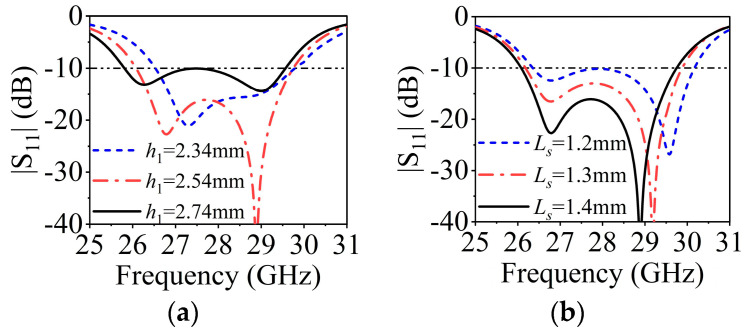
(**a**) Influence of *h*_2_ on |S_11_|. (**b**) Influence of *L_s_* on |S_11_|.

**Figure 10 sensors-23-04400-f010:**
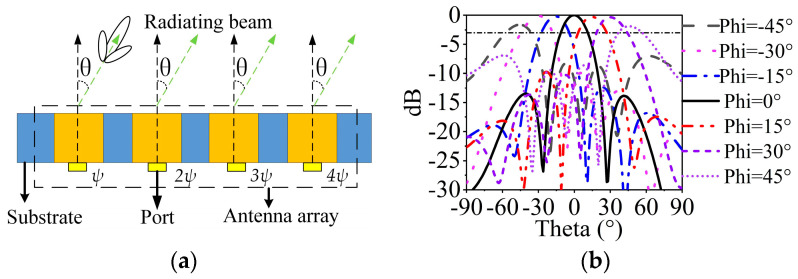
(**a**) Demonstration diagram of MMW beam scanning array. (**b**) Simulated MMW beam scanning performance at 28 GHz.

**Figure 11 sensors-23-04400-f011:**
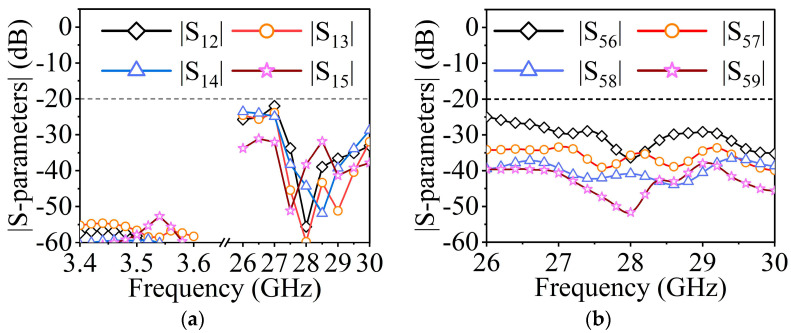
(**a**) Simulated isolations between port 1 and port 2–5. (**b**) Simulated isolations between port 5 and port 6–9.

**Figure 12 sensors-23-04400-f012:**
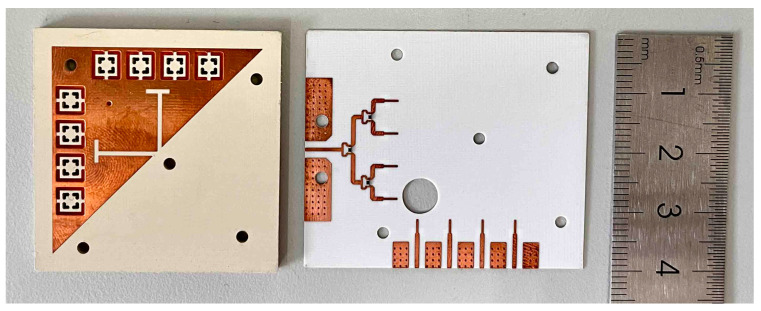
Photograph of the antenna prototypes.

**Figure 13 sensors-23-04400-f013:**
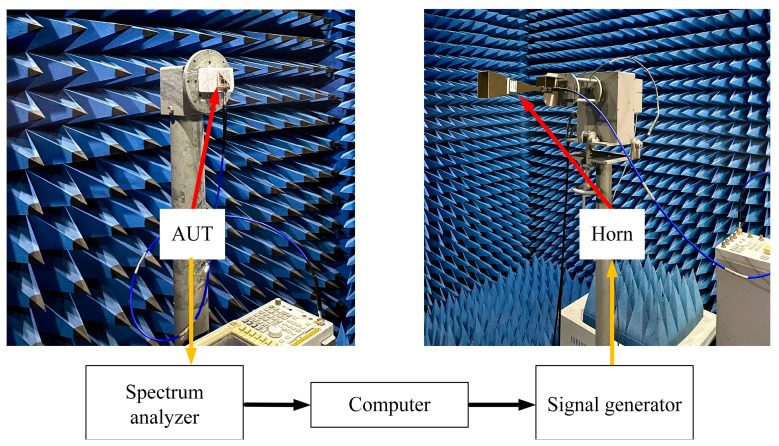
Measurement setup for radiation pattern.

**Figure 14 sensors-23-04400-f014:**
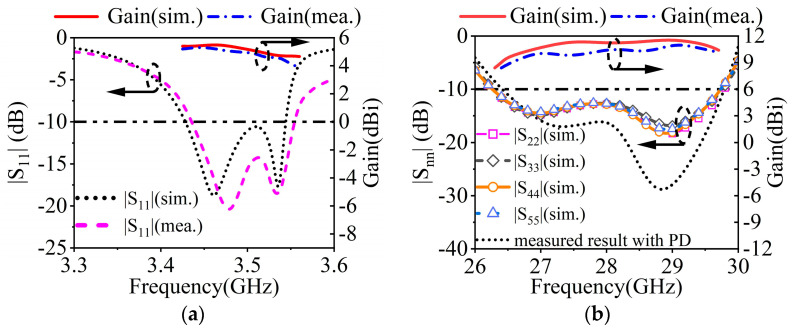
Simulated and measured |S_nn_|s and gains of the proposed antenna at (**a**) 3.5 GHz and (**b**) 28 GHz.

**Figure 15 sensors-23-04400-f015:**
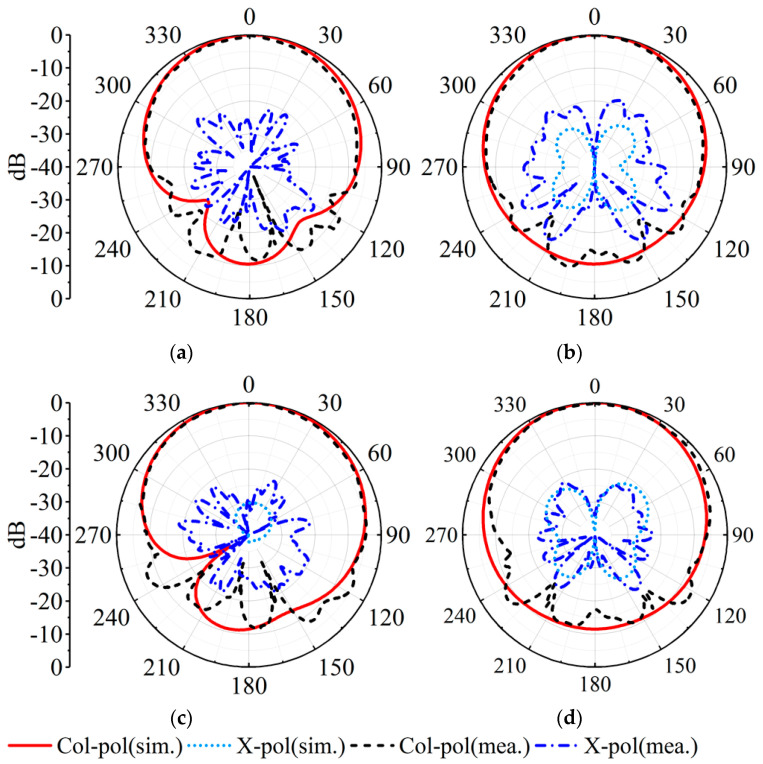
Simulated and measured realized gain patterns of the proposed MW antenna. (**a**) E-plane at 3.46 GHz. (**b**) H-plane at 3.46 GHz. (**c**) E-plane at 3.53 GHz. (**d**) H-plane at 3.53 GHz.

**Figure 16 sensors-23-04400-f016:**
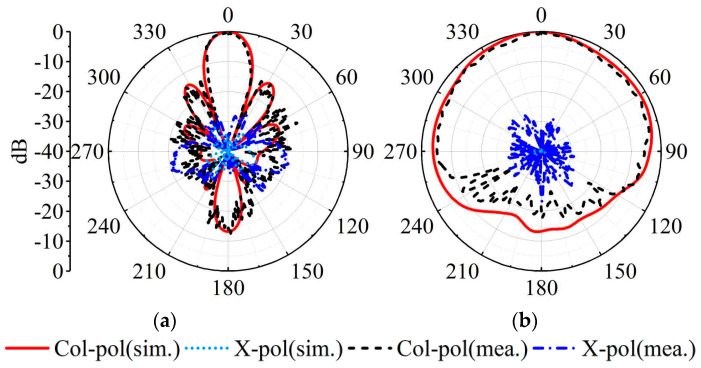
Simulated and measured realized gain patterns of the SIDRA array. (**a**) H-plane at 28 GHz. (**b**) E-plane at 28 GHz.

**Table 1 sensors-23-04400-t001:** Performance comparison with the previous antenna designs.

Ref.	Frequency (GHz)	Relative BW (%)	Peak Gain (dBi)	Size ($\lambda_{01}^{2}$)	Profile (*λ*_01_)	MMW Beam-Steering	MMW Dual Polarization
[9]	2.4/60	6.3/3.5	8.0/27.8	0.29	0.19	No	Yes
[10]	2.4/5/60	5.7/23.4/22.6	9.8/7.9/8.4	1.25	0.1	No	No
[17]	0.85/28	21/16	-/12.6	0.08	0.017	±37° (End-fife)	Yes
[20]	3.5/28	50.31/33.91	10.67/14.85	1.96	0.26	±20° (Broadside)	Yes
[22]	3.5/28	20.7/20.5	7.07/11.31	0.02	0.003	±25° (End-fife)	No
[24]	3.5/60	2.6/6.4	7.3/24	0.13	0.02	No	No
[28]	5.4/25	3.6/16.0	15.5/22.4	5.76	0.54	No	No
[29]	3.5/25.8	6/20	13.7/27.65	18.60	0.05	No	No
[30]	10/28	4.5/9.6	13.8/23.6	8.00	0.57	No	Yes
Prop.	3.5/28	3.4/11.8	5.34/11.0	0.08	0.03	±45° (Broadside)	Yes

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
