# Peer review of "A Compact Aperture-Sharing Sub-6 GHz/Millimeter-Wave Dual-Band Antenna"

_sensors, 2023, doi:10.3390/s23094400_

Round 1

Reviewer 1 Report

- In Figure 3(b), the authors present the mode analysis for 3-9GHz. The authors should explain it for M/M band also.

- What kind of S/W is used in this simulation ? 

- There are some differences in the X-pol pattern. The authors should explain it.

- What is the meaning of the X-pol pattern in Figure 12(b) ?

- The authors should present the measurement setup for radiation pattern.

Reviewer 2 Report

This article proposed, simulated, and measured a compact aperture-sharing antenna operating at 3.5 and 28 GHz band with dual-polarization and beam steering in MMW frequency band. The paper is well organized and can be accepted after minor modification listed as follows:

1-What do you mean of “Impedance band width”  in Table 1. Do you mean relative bandwidth? or fractional bandwidth ? Modify this issue.

2-In fig 9 it seem that  Wilkinson power dividers are used but lumped resistors are  not applied in divider structure add explanations about divider and applied resistors. 

3-Mutual coupling reduction should be considered  and briefly discussed in the introduction.

4-Design process flowchart should be added. And also explain how the Configuration of the proposed antenna , which depicted in Fig.2 and applied dimensions are obtained.

5-Many parts of paper are copied from [R1], which should be modified.

For example:

-caption of Fig.11.   

-caption of Fig.12.  

- and so on.

[R1] Wen-Wen Yang, Xin-Hao Ding, Tian-Wen Chen, Lei Guo, Wei Qin, Jian-Xin Chen. "A Sharedaperture Antenna for (3.5, 28) GHz Terminals with End-fire and Broadside Steerable Beams in Millimeter Wave Band", IEEE Transactions on Antennas and Propagation, 2022

Reviewer 3 Report

Authors have presented manuscript titled “A Compact Aperture-Sharing Sub-6 GHz/Millimeter-wave Dual-band Antenna”. Work is interesting and is supported with simulate and measured results. Following suggestions will be helpful to further improve the manuscript.

Introduction may be expanded further highlighting gaps in exiting work in literature. Consider expanding paragraph 1 and 2 only and comment on some specific papers.

Include all geometrical directions as many dimensions are missing.

Are there any air gaps between the layers. If not please specify.

For better understanding of the readers, add more explanation regarding modes.

The movement of beam is caused by phase difference among the elements, which is beam tilt. Add explanation on this and use beam tilt. It would be good to add simulated 3D patterns to demonstrate beam tilting for better understanding.

Expand the discussion in section 4 with more analysis insight.

Overall, the work is interesting and useful.

Round 2

Reviewer 1 Report

The authors responded appropriately to the reviewer's comments and revised well.

Reviewer 3 Report

Authors have addressed reviewers comments and the manuscript has been improved.